# Genetic Diversity of Avian Influenza Viruses Detected in Waterbirds in Northeast Italy Using Two Different Sampling Strategies

**DOI:** 10.3390/ani14071018

**Published:** 2024-03-27

**Authors:** Giulia Graziosi, Caterina Lupini, Federica Gobbo, Bianca Zecchin, Giulia Quaglia, Sara Pedrazzoli, Gabriele Lizzi, Geremia Dosa, Gabriella Martini, Calogero Terregino, Elena Catelli

**Affiliations:** 1Department of Veterinary Medical Sciences, University of Bologna, 40064 Ozzano dell’Emilia, BO, Italy; caterina.lupini@unibo.it (C.L.); giulia.quaglia2@unibo.it (G.Q.); sara.pedrazzoli2@unibo.it (S.P.); gabriele.lizzi2@unibo.it (G.L.); elena.catelli@unibo.it (E.C.); 2Comparative Biomedical Sciences Division, Istituto Zooprofilattico Sperimentale delle Venezie, 35020 Legnaro, PD, Italy; fgobbo@izsvenezie.it (F.G.); bzecchin@izsvenezie.it (B.Z.); cterregino@izsvenezie.it (C.T.); 3Veterinary Services, Local Health Unit of Imola (A.U.S.L. di Imola), 40026 Imola, BO, Italy; g.dosa@ausl.imola.bo.it (G.D.); ga.martini@hotmail.it (G.M.)

**Keywords:** avian influenza, wild birds, wetlands, whole-genome sequencing, cloacal swabs, avian faecal droppings

## Abstract

**Simple Summary:**

Avian influenza viruses (AIVs) significantly threaten the global poultry industry, and some strains are capable of infecting mammals, even humans. Wild aquatic migratory birds serve as the natural reservoir for low-pathogenic AIVs and, through migratory flyways, contribute to the global spread and genetic diversification of the virus. It is therefore essential to monitor AIVs circulating in wild hosts. In this study, the collection of cloacal swabs from waterbirds and avian faecal droppings in selected wetlands allowed for the molecular detection and characterization of different low-pathogenic AIV strains circulating in wild birds during the 2021–2022 avian influenza epidemic in the Italian poultry industry.

**Abstract:**

Avian influenza viruses (AIVs), which circulate endemically in wild aquatic birds, pose a significant threat to poultry and raise concerns for their zoonotic potential. From August 2021 to April 2022, a multi-site cross-sectional study involving active AIV epidemiological monitoring was conducted in wetlands of the Emilia-Romagna region, northern Italy, adjacent to densely populated poultry areas. A total of 129 cloacal swab samples (CSs) and 407 avian faecal droppings samples (FDs) were collected, with 7 CSs (5.4%) and 4 FDs (1%) testing positive for the AIV matrix gene through rRT-PCR. A COI-barcoding protocol was applied to recognize the species of origin of AIV-positive FDs. Multiple low-pathogenic AIV subtypes were identified, and five of these were isolated, including an H5N3, an H1N1, and three H9N2 in wild ducks. Following whole-genome sequencing, phylogenetic analyses of the hereby obtained strains showed close genetic relationships with AIVs detected in countries along the Black Sea/Mediterranean migratory flyway. Notably, none of the analyzed gene segments were genetically related to HPAI H5N1 viruses of clade 2.3.4.4b isolated from Italian poultry during the concurrent 2021–2022 epidemic. Overall, the detected AIV genetic diversity emphasizes the necessity for ongoing monitoring in wild hosts using diverse sampling strategies and whole-genome sequencing.

## 1. Introduction

Avian influenza virus (AIV, family *Orthomyxoviridae*, genus *Alphainfluenzavirus*) [1] infects birds and, occasionally, mammals, including humans, posing a great risk for animal and public health [2]. Wild aquatic birds of the Anseriformes and Charadriiformes orders (ducks, geese, gulls, and shorebirds) serve as the natural reservoir for low-pathogenic (LP) AIVs [3]. Through flyways, AIV-infected migratory birds facilitate the global spread of the virus and contribute to its genetic evolution and genomic reassortment [4,5,6]. In poultry, AIV clinical outcomes vary based on the host and virus-related factors. Gallinaceous birds typically exhibit mild or no disease when infected with LPAIVs [7]. However, if H5 or H7 AIV subtypes are involved, the virus can potentially evolve into highly pathogenic (HP) AI and cause severe outbreaks with high mortality rates in both wild birds and poultry [8,9]. Since the first detection in 1996 in domestic geese in China [10], HPAIVs belonging to the A/goose/Guangdong/1/1996 (Gs/Gd) (H5N1) genetic lineage have found their way back to their natural reservoir, the wild waterfowl, and have exhibited unprecedented global spread, genetic drift, and a tendency to reassort with LPAIVs [11,12,13]. Over time, the HPAIVs of H5 clade 2.3.4.4 have become dominant, and various subclades (a–h) have been classified according to the World Health Organization [14]. In October 2020, a novel H5N8 HPAIV of clade 2.3.4.4b first detected in the Netherlands in wild mute swans [15] spread throughout Europe, causing the most extensive and severe AI epizootic ever recorded in wild birds, poultry, and captive birds [16,17]. In Italy, between October 2021 and April 2022, 317 outbreaks of H5N1 HPAIVs of clade 2.3.4.4b were reported, resulting in the culling of over 8 million birds of all poultry categories [17]. Most of these outbreaks were concentrated in the northeastern part of the country, where densely populated poultry areas (DPPAs) are situated nearby wetlands, posing a high risk of AIV introduction via migratory aquatic birds [18]. Recent detections of HPAIVs in healthy migratory and resident wild ducks suggest a potential enzootic circulation of the virus in wild populations [19]. Field and experimental studies have pinpointed Eurasian wigeons (*Mareca penelope*), mallards (*Anas platyrhynchos*), and common teals (*Anas crecca*) as the most suitable long-distance carriers of HPAIVs along migratory flyways [18,20,21]. Cases of HPAIV infections in wild and domestic mammals are also being reported, posing a further potential risk of AI transmission to humans [22,23,24]. Additionally, global human population growth and natural habitat loss have created new opportunities for spillover events at the wildlife–domestic animal–human interface [25]. Therefore, understanding AIV circulation patterns in local ecosystems is crucial, providing fundamental information for AIV risk assessment and identifying potential hotspots for the emergence of novel strains [26].

This cross-sectional study hereby investigated the occurrence of AIVs in hunted or found dead wild birds and avian faecal droppings collected in wetlands located in the Emilia-Romagna region, northern Italy, during the 2021–2022 epidemic waves of H5 HPAI in poultry. The study area, characterized by DPPAs adjacent to critical wetlands for waterfowl migrating along the Black Sea/Mediterranean flyway, was selected for AIV monitoring using a combination of diverse sampling strategies. The detected AIVs were isolated and characterized utilizing whole-genome sequencing in order to perform sequence analysis and investigate phylogenetic relationships with previously detected isolates.

## 2. Materials and Methods

### 2.1. Background

The sampling sites comprised two inland wetlands and one coastal wetland, all situated in the Emilia-Romagna region, located at a maximum distance of 50 km from each other (Figure 1). These were chosen due to the presence of numerous overwintering and breeding waterfowl species migrating along the Black Sea/Mediterranean flyway that congregate and intermingle with resident populations of wild birds. Specifically, the “Campotto-Bassarone wetland” (44.589983, 11.826349) is a 600-hectare inland artificial freshwater impoundment at the confluence of the Idice and Reno rivers, located within the Delta Po Regional Park in Ferrara province. It is considered to be one of most important areas for aquatic migratory, breeding, and resident birdlife in Italy. It was designated as a Wetland of International Importance under the Ramsar Convention in 1949 and is part of the Site of Community Importance “Valli di Argenta” within the European Natura 2000 Network. The “Boscoforte peninsula” (44.550177, 12.154855) is a coastal wetland also within the Delta Po Regional Park in Ferrara province. This area is densely populated by migratory and resident shorebirds due to its proximity to the Adriatic Sea. It was chosen specifically to tentatively broaden the range of AIV hosts sampled, considering that shorebirds are often underrepresented in AIV monitoring activities in Europe [27,28]. Additionally, a private waterfowl hunting preserve in Bologna province was included in the study as a third location. Avian faecal dropping samples (FDs) were collected from “Campotto-Bassarone wetland”, “Boscoforte peninsula”, and the private hunting ground from August 2021 to April 2022, while cloacal swabs (CSs) were obtained from October 2021 to January 2022 from hunted or found dead wild birds in the hunting preserve or within the Delta Po Regional Park.

Access to all sampling locations was granted by relevant authorities or owners. The wild aquatic birds hunted or found dead included in this study were sampled within the application of the National AI Surveillance Plan 2021 (https://www.izsvenezie.it/documenti/temi/influenza-aviaria/piani-sorveglianza/piano-nazionale-influenza-aviaria-2021.pdf, accessed on 17 October 2021) and the Commission Delegated Regulation (EU) 2020/689. AI surveillance activities were carried out on behalf of the Veterinary Services of the Local Health Unit of Imola (A.U.S.L. of Imola) by sampling birds provided by local hunters or found dead by local ornithologists. Waterfowl species were hunted according to the National Hunting law 157/1992, without the necessity of any additional permits. While oropharyngeal swabs were submitted to institutes of the Italian National Health Service for concomitant HPAI testing, following the National AI Surveillance Plan, cloacal swabs were retained for this study.

### 2.2. Faecal Dropping Samples Collection

To estimate the positivity rate of AIVs during different phases of the bird migration phenology, samplings were conducted every two weeks during the southward autumn migration (August–October) and upward spring migration (February–April). Additionally, samples were collected once a month during the remaining months, totaling 15 field trips. The collection of fresh FDs was carried out at roosting sites of waterbirds (e.g., ducks, ibises, geese, flamingos, gulls, herons, shorebirds), identified with the assistance of local ornithologists, along 5 to 10 m in length on banks or scattered islets depending on the location. A total of 140 FDs were collected in wetlands of the private hunting preserve; there were 193 FDs in the “Campotto-Bassarone wetland” and 74 FDs in the “Boscoforte peninsula”. The avian species of origin of the FDs was determined through direct observation and/or photographic records of birds in the act of defecation. When identification by direct observation was not possible, the phenotypic characteristics of the faecal samples were utilized to recognize the bird family, without species-level identification. FDs were collected with a minimum distance of 15 cm from one to another to avoid sampling the same bird more than once. Samples were collected in duplicate using sterile rayon-tipped wooden swabs (Kaltek, Saonara, Italy). The portion of the FD with visible uric acids was excluded and, whenever feasible, only the upper-central portion of the faeces was collected to prevent potential contamination by PCR inhibitors from the soil. The swabs were then immersed in a sterile tube containing 1.0 mL of PBS/glycerol (9/1) buffer (Sigma-Aldrich, St. Louis, MO, USA), supplemented with antibiotics and antimycotic agents (10,000 IU/mL penicillin, 10 mg/mL streptomycin, 50 mg/mL gentamycin, and 10,000 IU/mL nystatin) (Sigma-Aldrich, St. Louis, MO, USA). FDs were kept on ice during the sampling activities and transportation, and then they were stored at −80 °C until analysis.

### 2.3. Cloacal Swab Samples Collection

Due to the unpredictable number of water birds hunted or found dead, an opportunistic sampling approach without a predefined sample size estimation was set. During 2021, a total of 31 birds were sampled in October, 48 were sampled in November, and 23 were sampled in December. Lastly, 17 birds were sampled in January 2022. For each sampled bird, information on species, sex, and age classes (juvenile of the year or adult) was recorded. Cloacal swabs were collected using a sterile rayon-tipped wooden swab and then they were immersed in 600 μL of PBS/glycerol buffer supplemented with antibiotics and antimycotic agents, as previously detailed. Samples were kept on ice during the sampling activities and transportation, and then they were stored at −80 °C until analysis.

### 2.4. Statistical Analysis

Differences in AIV molecular detections in the sampled hunted or found dead bird population were assessed using Fisher’s exact test. The “fisher.test” function in R software (version 4.0.4) [29] was employed, with a significance level set at 0.05. Comparisons were conducted separately for the categorical variables tested (male and female; adult and juvenile).

### 2.5. Sample Processing and Molecular Detection of AIVs

FDs were pre-processed to increase nucleic acid yields. Avian faeces have been described as challenging matrices for nucleic acids extraction due to the presence of molecules that can interfere with the procedure, such as uric acid, bile salts, complex polysaccharides, and nucleases [30,31]. After thawing on ice, FDs were vigorously mixed using a vortex mixer for an average of one minute per sample, or until the faecal pellet was homogenously resuspended. The samples were then pooled per sampling date and, whenever possible, per bird taxa, in groups of five, by aliquoting 100 μL/sample into a new tube. The pooled FD was further diluted in 1.0 mL of PBS. Similarly, CSs were thawed on ice, vortexed for one minute, and pooled in groups of five per sampling date and per species by aliquoting 100 μL/sample into a new 1.5 mL tube. FD and CS pools were then centrifuged at 11,000× *g* for 30 s at 4 °C.

RNA extraction was performed on 100 μL of supernatant from each pool using NucleoSpin^®^ RNA (MACHEREY-NAGEL, Düren, Germany) following the manufacturer’s instructions for cultured cells and tissue. Intype^®^ IC-RNA (INDICAL BIOSCIENCE GmbH, Leipzig, Germany), an exogenous RNA template, was employed as an internal control for RNA extraction. A multiplex real-time RT-PCR (rRT-PCR) protocol was applied for the detection of the matrix (M) gene of AIV [32] and the RNA template. rRT-PCR assays were performed in triplicate. The QuantiTect^®^ Multiplex RT-PCR kit (QIAGEN^®^, Hilden, Germany) was used on a Corbett Research Rotor-Gene™ RG-3000 platform (QIAGEN^®^, Hilden, Germany). Pools positive for AIVs were traced back to the original samples, which underwent individual RNA extraction and subsequent multiplex rRT-PCR analysis. For HA typing, rRT-PCRs targeting H5, H7, and H9 were applied [33,34,35]. A conventional one-step RT-PCR was further used to pathotype the hemagglutinin cleavage site by Sanger sequencing [36]. For other HA typing and for NA typing, rRT-PCRs were conducted using multiple oligonucleotide sets [37,38].

### 2.6. Virus Isolation in Specific Pathogen Free (SPF) Embryonated Eggs

To assess the viability of the detected AIVs, samples that tested positive for viral molecular detection were inoculated in 9-to-11-day-old SPF embryonated chicken eggs according to the standard methods [39,40]. For FDs, whenever possible, virus isolation was attempted using the duplicate sample that had not been previously thawed for molecular biology analyses. In brief, 500 μL of FD and CS supernatants was filtered through 0.22 μm Millipore filters and diluted in 1 mL of PBS with antibiotics and antimycotic agents (10,000 IU/mL penicillin, 10 mg/mL streptomycin, 50 mg/mL gentamycin, and 10,000 IU/mL nystatin). After centrifugation at 8000 rpm for 10 min, 200 μL of the supernatant was inoculated into the allantoic cavity of the SPF eggs (five eggs per sample). The eggs were incubated at 37 °C until embryo death was observed or up to seven days. Inoculated SPF eggs were then chilled for a minimum of four hours; the harvested allantoic fluids were submitted to a hemagglutination assay (HA) and hemagglutination inhibition (HI) testing.

### 2.7. Genome Sequencing and Phylogenetic Analysis

The obtained AIV isolates underwent further RNA extraction using the NucleoSpin^®^ RNA (MACHEREY-NAGEL, Düren, Germany) following the manufacturer’s instructions for cultured cells and tissue. The extracted RNA was subjected to whole-genome sequencing using a Next-Generation Sequencing (NGS) technology on the Illumina MiSeq System (Illumina Inc., San Diego, CA, USA). The preparation of libraries, the assessment of their quality and quantity, and sequencing reactions were conducted as described elsewhere [18].

For phylogenetic analyses, the viral sequences were aligned against and compared with previously published AIV sequences available on the Global Initiative on Sharing All Influenza Data (GISAID) Epi-Flu Database (https://platform.epicov.org/, accessed on 13 December 2023) and GenBank (https://www.ncbi.nlm.nih.gov/genbank/, accessed on 13 December 2023). Viruses showing high sequence identity to the isolates hereby obtained were included in the analysis. Multiple sequence alignment was performed using the Multiple Alignment with Fast Fourier Transformation (MAFFT) online service with default parameters [41]. Alignments were manually trimmed in Geneious Prime^®^ v.2022.2.2, and the gene segment lengths used for phylogenetic analysis were as follows: PB2 (2277 nt), PB1 (2271 nt), PA (2148), HA (H1 subtype) (1648 nt), HA (H5 subtype) (1645 nt), HA (H9 subtype) (1627 nt), NP (1495 nt), NA (N1 subtype) (1336 nt), NA (N2 subtype) (1408 nt), NA (N3 subtype) (1408 nt), M (1013), and NS (890). The best partition scheme, substitution model selection based on the Bayesian information criterion (BIC), and maximum likelihood phylogenetic reconstruction were performed on the IQ-TREE web server [42,43] separately for each gene (PB2, PB1, PA, HA, NP, NA, M, NS). The robustness of the inferred clades was evaluated using 1000 ultrafast bootstrap (UFBoot) replicates. The phylogenetic tree was visualized by using FigTree v.1.4.2.

### 2.8. Sequence Analysis

The full-length sequences of all eight AIV gene segments (PB2, PB1, PA, HA, NP, NA, M, and NS) were analyzed to assess the presence of molecular markers associated with mammalian host adaptation, pathogenicity, and drug resistance. The sequences were submitted to the FluSurver mutations app of the GISAID Initiative [44] and manually screened based on inventories of markers affecting AIV biological properties [45,46]. The identified amino acid substitutions are referred to using H3 numbering.

Nucleotide similarity searches were performed on the GISAID Epi-Flu database using Basic Local Alignment Search Tool (BLAST nucleotide—BLASTn) to compare the gene segments obtained in this study with those of previously published strains. Following MAFFT alignment, amino acid substitutions of the HA and NA gene segments of the H1N1, H5N3 LPAI, and H9N2 strains were annotated in comparison to their closest relatives.

### 2.9. Cytochrome Oxidase I (COI) Barcoding of Faecal Dropping Samples for Host Identification

FDs that tested positive for AIV molecular detection were analyzed for the identification or confirmation of the avian species of origin using a PCR targeting the cytochrome c oxidase subunit I (COI) mitochondrial DNA [47]. After thawing the sample on ice, the faecal pellet was resuspended into PBS/glycerol buffer using a vortex mixer. Genomic DNA from the faecal suspension was extracted using the QIAamp^®^ Fast DNA Stool mini kit (QIAGEN^®^, Hilden, Germany) following the manufacturer’s instructions and eluted in a final volume of 100 μL. A 749 bp region of the COI gene was amplified by PCR using primers BirdF1 (5′-TTCTCCAACCACAAAGACATTGGCAC-3′) and BirdR1 (5′-ACGTGGGAGATAATTCCAAATCCTG-3′) [48]. PCR products were purified using ExoSAP-IT™ PCR Product Cleanup Reagent (Applied Biosystems, Waltham, MA, USA), and sequencing was performed by a commercial sequencing service (Macrogen Europe, Amsterdam, The Netherlands). Nucleotide sequences were edited and assembled using Geneious Prime^®^ v.2022.2.2 and then queried against the Barcode of Life Data System database (http://boldsystems.org, accessed on 19 October 2023) to attempt taxonomic species identification [49].

## 3. Results

### 3.1. Wild Bird Population Sampled

Overall, 536 samples were collected for successive AIV molecular detection. Of these, 407 were FDs collected between August 2021 and April 2022. As identified through the direct or photographic observation of birds in the act of defecation, the majority of the FDs belonged to ducks (249/407, 61.2%), followed by geese (46/407, 11.3%), shorebirds (34/407, 8.3%), herons (30/407, 7.3%), gulls (19/407, 4.6%), ibises (18/407, 4.4%), rails (10/407, 2.4%), and flamingos (1/407, 0.2%) (Figure 2A). A total of 129 CSs were obtained from waterbirds hunted or found dead from October 2021 to January 2022. A total of 85 Eurasian teals (*A. crecca*), 22 Northern shovelers (*Spatula clypeata*), 8 mallards (*A. platyrhynchos*), 5 Northern lapwings (*Vanellus vanellus*), 3 greylag geese (*Anser anser*), 2 gadwalls (*Mareca strepera*), 2 common snipes (*Gallinago gallinago*), 1 Eurasian wigeon (*M. penelope*), and 1 spotted redshank (*Tringa erythropus*) were sampled. Of these, 59.7% were identified as females (77/129), 38% (49/129) as males, and 2.3% (3) were undetermined (Figure 2B). Concerning age classes, 47.3% (61/129) of the birds were aged as juveniles (first calendar year) and 52.7% (68/129) as adults (Figure 2C).

### 3.2. AIVs Detection in Faecal Droppings and Cloacal Swabs

Overall, 11 out of 536 samples (2%) tested positive for AIV at rRT-PCR for M gene. Of these, four AIVs were detected in FDs (4/407, 1%) (Table 1) and seven in CSs (7/129, 5.4%) (Table 2). Positive samples were collected between October and November 2021 during the late autumn migration and wintering period. AIV detections were further subtyped, virus isolation was attempted, and FDs species of origin were successfully identified through COI barcoding.

With respect to FDs (Table 1), the following AIVs were detected through molecular testing: one H5N3 LPAIV and one HxN9 in wild mallards (*A. platyrhynchos*) sampled in the “Campotto-Bassarone wetland”; one H9N2 in a greylag goose (*A. anser*) sampled in the “Campotto-Bassarone wetland”; and one HxN2 in a common snipe (*G. gallinago*) sampled in the “Boscoforte peninsula”. FDs of ibises, gulls, herons, flamingos, and rails did not yield positive results for AIV molecular detection. Among AIVs from FDs, only the H5N3 LPAIV strain was successfully isolated (A/mallard/Italy/22VIR4203-2/2021 (H5N3)) and subsequently subjected to NGS.

With respect to CSs (Table 2), the following AIVs subtypes were molecularly identified in Eurasian teals (*A. crecca*): H1N1 (in one juvenile female), HxN9 (in one juvenile male), H9N2 (in one juvenile male and three juvenile females), and H6N1 (in one adult male). Of these, four viruses were successfully isolated and subjected to NGS, namely A/teal/Italy/1821-10_22VIR4622-1/2021 (H1N1), A/teal/Italy/1856-7_22VIR4622-7/2021 (H9N2), A/teal/Italy/1828-6_22VIR4622-5/2021 (H9N2), and A/teal/Italy/1821-14_22VIR4622-3/2021 (H9N2). The full-length AIVs viral sequences obtained by NGS were submitted to the GISAID EpiFlu Database (Appendix A).

The results of Fisher’s exact tests showed a non-significant association between sex (*p*-value = 1) or age classes (*p*-value= 0.05184) and AIV molecular detection.

None of the oropharyngeal swabs submitted for HPAI testing yielded a positive result.

### 3.3. Sequence Analysis

The HA cleavage site of the H5N3 strain (amino acid residues at positions 321–329: PQRETR↓GLF) did not contain multi-basic amino acids (arginine, R, or lysine, K); therefore, the virus was classified as LPAI. Full-length sequences analysis of the three isolated H9N2 AIVs showed a 100% identity among them, and a cleavage site motif of low pathogenicity for chickens (amino acid residues at positions 324–332: PAASDR↓GLF) [50,51,52].

Experimentally verified molecular markers associated with putative enhanced AIV adaptation to mammals, pathogenicity, and drug resistance were identified in the H1N1, H5N3, and H9N2 viruses (Table 3).

The results of the BLAST nucleotide sequence similarity searches, based on sequences available on the GISAID Epi-Flu Database, are summarized in Table 4. The H1N1, H5N3 LPAIV, and the three H9N2 isolates hereby obtained shared 98–100% nucleotide sequence identity with AIVs previously detected in Eurasia. Specifically, the PB2, PB1, PA, HA, NP, and NA gene segments of the H1N1 isolate exhibited high nucleotide similarity (98–100%) to H1N1, H4N6, and H5N2 LPAIVs isolated in Northern Europe (2017–2021). Additionally, the M and NS genes shared a 99–100% nucleotide sequence similarity with H3N2 and H10N3 viruses isolated in Russia and the People’s Republic of China in 2021 and 2019, respectively. All gene segments of the H5N3 LPAIV strain showed the highest nucleotide sequence similarity (99%) to other H5N3 LPAIVs isolated in wild birds in Italy during 2021. Regarding the H9N2 viruses, PB2, PB1, HA, NP, NA, and NS genes exhibited the highest similarities with LPAIVs detected in European countries during 2015–2020 (Table 4), while the PA gene segment displayed a 98% nucleotide sequence similarity with a homologous sequence of H3N8 isolated from wild birds in western Siberia (2020). Lastly, the M gene segment showed the highest genetic similarity (98%) to an H3N6 AIV strain isolated in Egypt in 2015.

H1N1, H5N3, and the H9N2 viruses hereby isolated the presented specific amino acid substitutions in HA and NA segments in comparison to their closest relatives (Table 5).

### 3.4. Phylogenetic Analyses

Phylogenetic analyses of the HA and NA gene sequences revealed that the viruses isolated in the current study were closely related to AIVs circulating in wild birds or, to a lesser extent, in domestic birds in Europe during 2020–2022. The H1 hemagglutinin gene of A/teal/Italy/1821-10_22VIR4622-1/2021 (H1N1) was in a sister group relationship with another H1N1 isolate from Belgium (UFBoot, 100), and these two viruses formed a monophyletic clade (UFBoot 99) with A/mallard/Netherlands/18015513-001/2018 (H1N1) (Figure 3).

The H5 HA gene of A/mallard/Italy/22VIR4203-2/2021 (H5N3) LPAI was in a sister group relationship (UFBoot 64) with A/mallard/Netherlands/7/2022, and these were included in a monophyletic clade with other H5 LPAI isolates from the Netherlands and Italy obtained from wild birds during 2021–2022 (Figure 4).

The H9 viruses obtained in this study clustered within the Y439-like sub-lineage in the Eurasian lineage of H9 hemagglutinin gene (Figure 5). These viruses formed a monophyletic clade with maximum support, which was in a sister relationship with an H9N2 isolate from a mallard in Poland obtained in 2020.

The N1 gene segment of the H1N1 isolate clustered together (UFBoot 98) with A/chicken/Denmark/S0275-3/2020 (H5N1) LPAIV isolated in chickens (Figure 6). The N3 NA gene of the H5N3 LPAIV was in a sister group relationship (UFBoot 72) with a clade composed of NA genes of H5N3 LPAI Italian isolates from wild ducks (Figure 7). The N2 NA gene segments of the three H9N2 isolates formed a maximum-supported monophyletic clade in a sister group relationship with a clade including N2 NA genes of German and Polish isolates of H5N2 HPAIVs from swans (A/swan/Germany-BW/AI00996/2022 and A/swan/Germany-BW/AI00997/2022) and from chickens (A/chicken/Poland/H182_22VIR2515-1/2022) (Figure 8).

The phylogenetic analyses of the PB2, PB1, PA, M, NS, and NP gene segments showed that, for each AIV subtype hereby detected, these genes arose from different ancestors, and the closest relatives were isolated both in Europe and in Asia (Appendix A).

## 4. Discussion

To actively monitor AIVs occurring in wild birds in northern Italy, where densely populated poultry areas are located, a cross-sectional study was conducted from August 2021 to April 2022. This survey included the collection and testing of cloacal swabs and avian faecal droppings for AIV molecular detection. A total of 11 LPAIVs were obtained, belonging to subtypes H1N1, H5N3, H6N1, H9N2, HxN9, and HxN2; 5 of these (one H1N1, one H5N3, and three H9N2 virus strains) were isolated and fully sequenced. HPAIVs were not detected despite the concurrent widespread circulation of HPAI H5N1 viruses of clade 2.3.4.4b in Italian poultry, linked to 317 outbreaks and over 8 million birds culled from October 2021 to April 2022 [17]. AI outbreaks were concentrated in poultry farms in the Po valley of the Veneto and Lombardy regions, northeast Italy, while the Emilia-Romagna region, where this study was conducted, reported only two HPAI cases in non-commercial poultry farms in Rimini and Ravenna province, near wetlands [65]. With respect to wild birds, during the 2021–2022 surveillance activities, a total of 23 positive cases of HPAIV of clade 2.3.4.4b infections were reported [66]. These were primarily found in scavenging or predator birds, such as raptors, owls, corvids, and gulls, indicating a wider host range than previous years [17].

Species belonging to the Anseriformes and Charadriiformes orders, specifically the mallard, greylag goose, and common snipe, as identified through COI-barcoding, tested positive for LPAIVs molecular detection in faecal droppings. FDs of shorebirds exhibited the highest positivity rate (3%, 1/34), followed by those of geese (2.1%, 1/46) and ducks (0.8%, 2/249). Nevertheless, the convenience sampling strategy employed might have resulted in an overestimation of the positivity rate for shorebirds and geese, as these groups were represented by smaller samples sizes compared to the ducks. While mallards and common snipes are listed as huntable according to Italian legislation (National Hunting Law 157/1992) and could be monitored in our study through cloacal swabbing, the majority of other bird groups sampled via FDs collection included non-huntable species. The use of an environmental sampling strategy in this study proved to be a valuable tool for broadening the host range in AIV monitoring. While active surveillance in wild birds is considered ideal, the utilization of avian faecal droppings from the environment may offer a less biased and cost-effective approach to AIV surveillance [67,68]. Both LPAI and HPAI viruses can indeed remain viable in bird faeces for weeks, even months, at low temperatures [69]. This non-invasive sampling strategy might be particularly valuable amid the current widespread circulation of HPAIVs H5N1 of clade 2.3.4.4b among different wild bird species [23,70]. Furthermore, strategically located wetlands, such as those included in this study near DPPAs, where numerous migratory birds feed and rest, could serve as potential sites for the emergence of novel AI strains. Consequently, these areas should be prioritized for year-long environmental monitoring. The successful detection of AIVs in faecal dropping samples, coupled with the identification of the bird species of origin through COI-barcoding, paves the way for employing this sampling strategy to enhance wild animal surveillance in the vicinity of poultry farms within HPAI restriction zones as a useful tool for monitoring potential virus spillover at the domestic bird–wildlife interface [71,72]. Environmental matrices, however, might yield lower viral loads than swabs collected from wild birds [73]. Particularly, the limited amount of faecal material, low AIV cloacal shedding, pH of the faeces, and the presence of inhibitory substances in faecal matter could potentially limit the chance of AIV detection, subtyping, and isolation in freshly collected FDs [73,74].

With respect to the sampling activity on hunted or found dead waterbirds, Eurasian teals constituted the majority (65.9%, 85/129) of birds tested for AIV molecular detection. An 8.2% LPAIV positivity rate was recorded for this species, aligning with previous studies on AIV molecular surveys conducted in teals where prevalence ranged from 3.6% to 15% [75,76,77,78,79]. Notably, Eurasian teals have been found positive for HPAIVs H5Nx of clade 2.3.4.4b, displaying asymptomatic infection associated with oropharyngeal and cloacal shedding [18,80,81]. Regarding the absence of AI viral detection in other species hereby sampled, this could be attributed to the low sample size obtained.

In this study, LPAIVs were exclusively detected in October and November 2021, in coincidence with waterbirds’ autumn migration and early wintering periods. No virus circulation was observed in the remaining months. However, during the spring migration period, results could have been biased due to the smaller sample size of FDs and the absence of cloacal swab collection due to the suspension of hunting activities. A higher AIV prevalence during autumn and early winter was also reported in surveys conducted on wild bird faecal droppings collected in wetlands in Spain and Mongolia [82,83]. During the autumn months, the congregation of birds from diverse origins, mingling with the local avian community, facilitates multiple-AIV strain transmission. This, coupled with the presence of immunologically naïve juveniles that amplify the infection, contributes to increased AIV prevalence in reservoir species [84,85,86]. The subsequent onset of homo- and heterosubtypic immunity likely protect birds against infection from phylogenetically close AIV strains, reducing viral circulation during following seasons [21,85,87].

Considering AI viruses’ high mutation rate and diverse gene constellations resulting from reassortment events in wild birds, whole-genome sequencing plays a pivotal role in characterizing the isolated strains through sequence and phylogenetic analyses [67]. Gene segments of the LPAIVs obtained in this study showed close genetic relationships with viruses isolated from wild birds, and, to a lesser extent, domestic birds in Europe, Northern Africa, and Asia. Strains identified within the same bird migratory flyway, such as the Black Sea/Mediterranean flyway that encompasses Italy, are indeed more similar to each other than those found elsewhere [88]. Interestingly, none of internal genes of the isolates obtained in this study clustered together with gene segments belonging to HPAIVs involved in the concurrent poultry epidemic occurring in northeast Italy. This allows us to speculate that the AIV genetic diversity found in our study primarily arose from reassortment events, which occurred in wild bird populations. Among the HA and NA gene segments of the strains hereby isolated, one to five amino acid substitutions were identified in comparison to AIVs with the highest nucleotide sequence similarity; however, none of these mutations was associated with molecular markers affecting viral biological properties [45,46]. Through sequence analysis, several other amino acid residues have been identified as potentially relevant within the context of AIV increased zoonotic risk. With respect to H5N3 LPAIV, the 159N amino acid residue identified in the HA gene segment has been linked to increased binding to mammal-type receptors (α2,6-linked sialic acid), as demonstrated for a live attenuated influenza A/Vietnam/1203/2004 (H5N1) vaccine virus in challenged ferrets [59]. This amino acid substitution has been sporadically observed in the H5 gene segments of HPAIVs circulating in birds in Europe from 2015 onwards [89,90,91]. The 155T amino acid residue in the H9 gene segment of the isolates hereby obtained has been reported to be linked to increased binding affinity for the mammal-like receptor in HPAIV H5N1 and H9N2 [57,58]. The HA-H9 sequences also carried the D225G amino acid substitution, which has been associated with a broader tissue tropism and increased replication in swine in an experimental study involving an H9N2-2009 pandemic H1N1 reassortant virus generated through reverse genetics [60]. Lastly, the A30T amino acid substitution found in the NA gene segment of the H9N2 viruses has been detected following airborne transmission in ferrets experimentally infected with H9N2-2009 pandemic H1N1 reassortant virus [61]. Overall, H9N2 represents the virus subtype more frequently detected in our study in both cloacal swabs and faecal droppings; this subtype has been shown to endemically circulate in poultry in Asia, the Middle East, and parts of Africa and has a high prevalence of infection in wild birds [92,93,94]. Given the H9N2 virus isolations also from swine and humans, these AIVs are considered to be potentially involved in the emergence of the next influenza pandemic and should therefore be closely monitored in birds [93,95].

## 5. Conclusions

The results of this study revealed the active circulation of multiple LPAIV subtypes in wild waterbirds in northern Italy. The survey involved the collection of cloacal swabs from waterbirds and avian faecal droppings from wetlands, thus detecting a broader range of AIV subtypes than each sampling strategy could have achieved individually. Despite the close genetic relationship with other common AIV isolates previously found in Eurasia and Northern Africa, the HA and NA gene segments of the viruses obtained in this study displayed several unique amino acid changes. Potential markers associated with increased AIV zoonotic risk were also identified, but the actual effect of these mutations on the biological characteristics of the obtained viruses is unknown. Further in vitro and in vivo experiments are needed in this regard to improve the existing knowledge. Wild waterfowl in Eurasia harbors a vast genetic pool of AIVs, contributing to the emergence of novel strains even capable of infecting new hosts, such as domestic poultry and mammals. The genetic diversity hereby detected emphasizes the need for continued surveillance in wild hosts using different sampling strategies and whole-genome sequencing characterization of the isolates.

## Figures and Tables

**Figure 1 animals-14-01018-f001:**
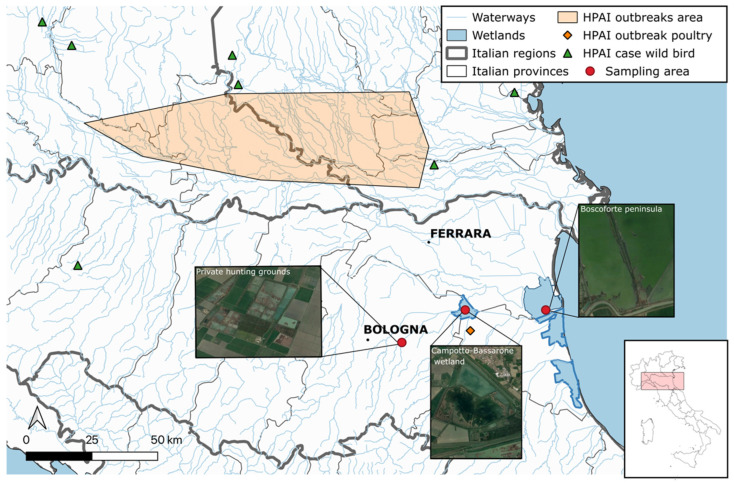
Sampling sites in the Emilia Romagna region, northern Italy, selected for investigating the occurrence of AIVs in hunted or found dead wild birds and avian faecal droppings. Outbreaks in poultry (orange area and orange diamond) and cases in wild birds during the 2021–2022 AI epidemic are reported. Created with QGIS v.3.6.0.

**Figure 2 animals-14-01018-f002:**
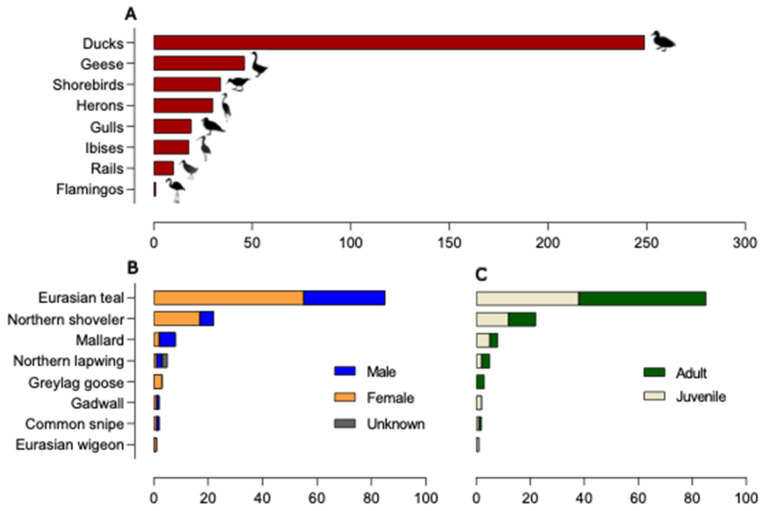
Wild bird population sampled. (**A**) Bird taxa/number of faecal droppings collected in wetlands; (**B**) species/number of hunted or found dead wild birds included in the study according to sex; (**C**) species/number of hunted or found dead wild birds included in the study according to age classes.

**Figure 3 animals-14-01018-f003:**
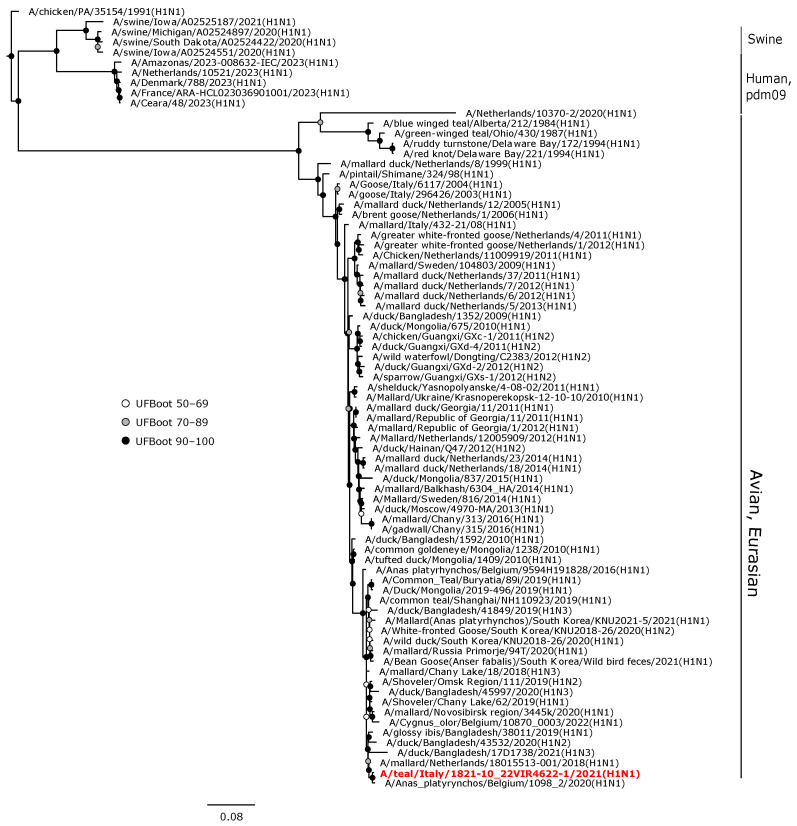
Phylogenetic analysis of the H1 of the A/teal/Italy/1821-10_22VIR4622-1/2021(H1N1) isolate. The assembled full H1 sequence hereby obtained (red, bold) is compared with full coding sequences available at GenBank and/or GISAID. Human isolates of the pandemic influenza A(H1N1)2009 virus lineage and swine A(H1N1) viruses were included. Circles at nodes indicate the ultrafast bootstrap (UFBoot) score range and are labeled in white (UFBoot 50–69), grey (UFBoot 70–89), and black (UFBoot 90–100). The substitution model for the maximum likelihood phylogenetic reconstruction was GTR+F+I+G4.

**Figure 4 animals-14-01018-f004:**
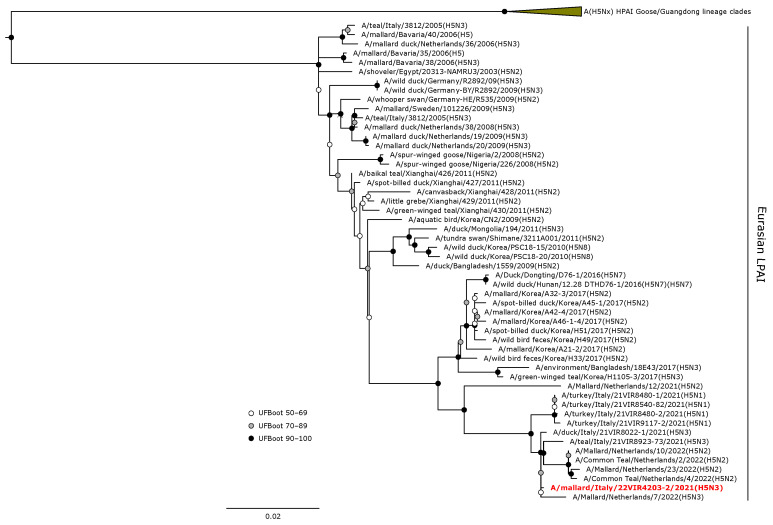
Phylogenetic analysis of the H5 of the A/mallard/Italy/22VIR4203-2/2021 (H5N3) LPAI isolate. The assembled full H5 sequence hereby obtained (red, bold) is compared with full coding sequences available at GenBank and/or GISAID. The collapsed clade in the upper part of the phylogenetic tree shows the HPAIVs isolated in Europe during 2020–2023. Circles at nodes indicate the ultrafast bootstrap (UFBoot) score range and are labeled in white (UFBoot 50–69), grey (UFBoot 70–89), and black (UFBoot 90–100). The substitution model for the maximum likelihood phylogenetic reconstruction was GTR+F+I+G4.

**Figure 5 animals-14-01018-f005:**
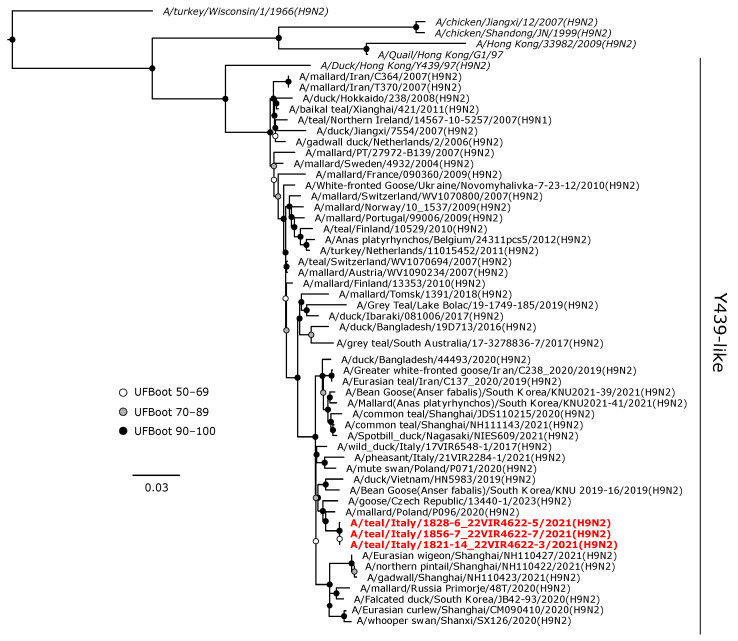
Phylogenetic analysis of the H9 of the A/teal/Italy/1856-7_22VIR4622-7/2021(H9N2), A/teal/Italy/1828-6_22VIR4622-5/2021(H9N2), and A/teal/Italy/1821-14_22VIR4622-3/2021(H9N2) isolates. The assembled full H9 sequences hereby obtained (red, bold) are compared with full coding sequences available at GenBank and/or GISAID. Reference sequences are shown in italics. Circles at nodes indicate the ultrafast bootstrap (UFBoot) score range and are labeled in white (UFBoot 50–69), grey (UFBoot 70–89), and black (UFBoot 90–100). The substitution model for the maximum likelihood phylogenetic reconstruction was GTR+F+G4.

**Figure 6 animals-14-01018-f006:**
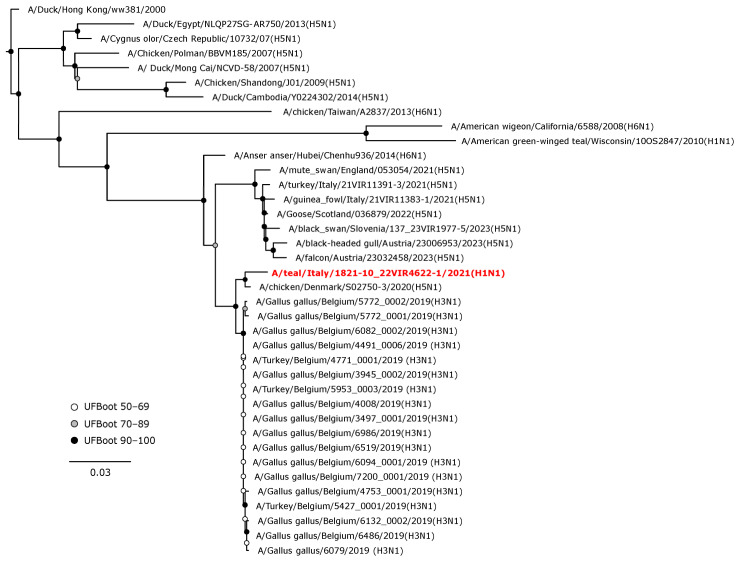
Phylogenetic analysis of the N1 of the A/teal/Italy/1821-10_22VIR4622-1/2021(H1N1) isolate. The assembled full N1 sequence hereby obtained (red, bold) is compared with full coding sequences available at GenBank and/or GISAID. Circles at nodes indicate the ultrafast bootstrap (UFBoot) score range and are labeled in white (UFBoot 50–69), grey (UFBoot 70–89), and black (UFBoot 90–100). The substitution model for the maximum likelihood phylogenetic reconstruction was TIM2+F+G4.

**Figure 7 animals-14-01018-f007:**
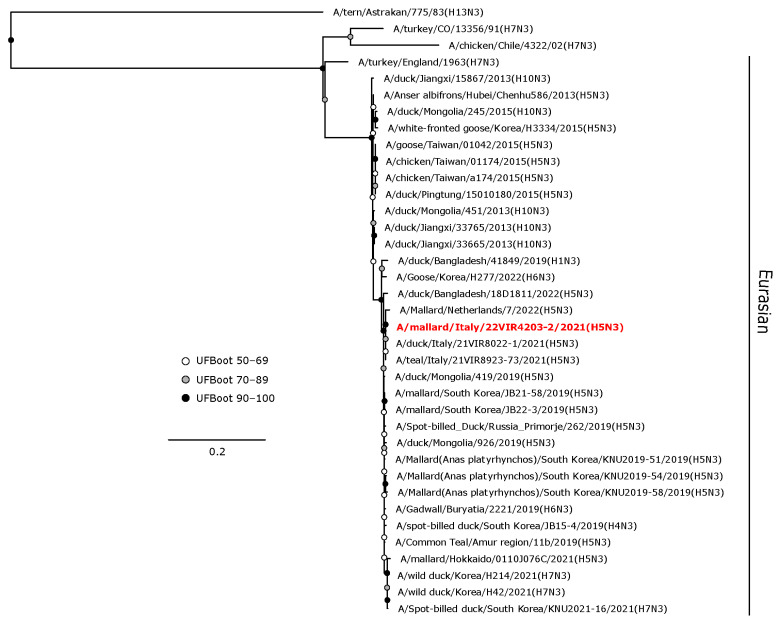
Phylogenetic analysis of the N3 of the A/mallard/Italy/22VIR4203-2/2021 (H5N3) LPAI isolate. The assembled full N3 sequence hereby obtained (red, bold) is compared with full coding sequences available at GenBank and/or GISAID. Circles at nodes indicate the ultrafast bootstrap (UFBoot) score range and are labeled in white (UFBoot 50–69), grey (UFBoot 70–89), and black (UFBoot 90–100). The substitution model for the maximum likelihood phylogenetic reconstruction was TIM2+F+G4.

**Figure 8 animals-14-01018-f008:**
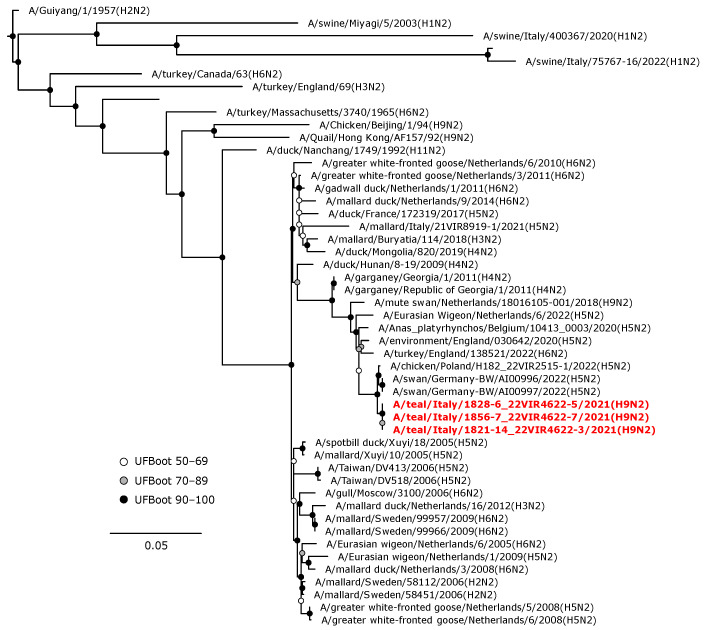
Phylogenetic analysis of the N2 of the A/teal/Italy/1856-7_22VIR4622-7/2021(H9N2), A/teal/Italy/1828-6_22VIR4622-5/2021(H9N2), and A/teal/Italy/1821-14_22VIR4622-3/2021(H9N2). The assembled full N2 sequences hereby obtained (red, bold) are compared with full coding sequences available at GenBank and/or GISAID. Circles at nodes indicate the ultrafast bootstrap (UFBoot) score range and are labeled in white (UFBoot 50–69), grey (UFBoot 70–89), and black (UFBoot 90–100). The substitution model for the maximum likelihood phylogenetic reconstruction was TVM+F+G4.

**Table 1 animals-14-01018-t001:** Sample size, positivity rate, and AIV subtypes identified in faecal dropping samples in three wetlands of the Emilia-Romagna region, northern Italy. Phases of the bird migration phenology and bird taxa of origin of the samples are detailed.

	AIV Positive/Sample Size (Positivity Rate %)	AIV Subtype
Bird migration phenology period		
Autumn migration ^†^	2/203 (1)	HxN9, H5N3
Wintering ^††^	2/134 (1.5)	H9N2, HxN2
Spring migration ^†††^	0/70 (0)	n.a. °
Bird taxa of origin		
Ducks	2/249 (0.8)	HxN9, H5N3
Geese	1/46 (2.1)	H9N2
Ibises	0/18 (0)	n.a.
Gulls	0/19 (0)	n.a.
Herons	0/30 (0)	n.a.
Flamingos	0/1 (0)	n.a.
Shorebirds	1/34 (3.0)	HxN2
Rails	0/10 (0)	n.a.
Total	4/407 (1.0)	HxN9, H5N3, H9N2, HxN2

^†^ August–October; ^††^ November–January; ^†††^ February–April. ° n.a., not applicable.

**Table 2 animals-14-01018-t002:** Sample size, positivity rate, and AIV subtypes identified in cloacal swabs in waterbirds hunted or found dead in wetlands of the Emilia-Romagna region, northern Italy.

Bird Species	AIV Positive/Sample Size (Positivity Rate %)	AIV Subtype
Anseriformes order		
Eurasian teal (*Anas crecca*)	7/85 (8.2)	H1N1, HxN9, H9N2, H6N1
Mallard (*Anas platyrhynchos*)	0/8 (0)	n.a. °
Greylag goose (*Anser anser*) ^†^	0/3 (0)	n.a.
Eurasian wigeon (*Mareca penelope*)	0/1 (0)	n.a.
Gadwall (*Mareca strepera*)	0/2 (0)	n.a.
Northern shoveler (*Spatula clypeata*)	0/22 (0)	n.a.
Charadriiformes order		
Common snipe (*Gallinago gallinago*)	0/2 (0)	n.a.
Spotted redshank (*Tringa erythropus*) ^†^	0/1 (0)	n.a.
Northern lapwing (*Vanellus vanellus*) ^†^	0/5 (0)	n.a.
Total	7/129 (5.4)	H1N1, HxN9, H9N2, H6N1

^†^ Individuals found dead. ° n.a., not applicable.

**Table 3 animals-14-01018-t003:** Amino acid (aa) sequence analysis of the five AIV strains hereby isolated for the identification of molecular markers potentially relevant for mammalian host adaptation, pathogenicity, and drug resistance. Experimentally verified markers in the polymerase basic protein 2 (PB2), polymerase basic protein 1 (PB1), polymerase (PA), hemagglutinin (HA), neuraminidase (NA), non-structural protein 1 (NS-1), and matrix 1 (M1) genes were identified and reported.

Viral Protein	Amino Acid Substitution	A/Teal/Italy/1821-10_22VIR4622-1/2021 (H1N1)	A/Mallard/Italy/22VIR4203-2/2021 (H5N3)	A/Teal/Italy/1856-7_22VIR4622-7/2021 (H9N2)	A/Teal/Italy/1828-6_22VIR4622-5/2021 (H9N2)	A/Teal/Italy/1821-14_22VIR4622-3/2021 (H9N2)	Phenotype (Subtype Tested)	Ref.
PB2	I292V ^a^	I	I	V	n.a. *	n.a. *	Increased polymerase activity in mammalian cell line, increased virulence in mice (H9N2)	[53]
	V598T/I	T	T	T	T	T	Increased polymerase activity and replication in mammalian cells, increased virulence in mice (H7N9)	[54]
PB1	D3V ^a^	V	V	V	V	V	Increased polymerase activity and viral replication in avian and mammalian cell lines (H5N1)	[55]
PA	S37A ^a^	A	A	A	A	A	Increased polymerase activity in mammalian cell line (H7N9)	[56]
HA	I155T ^b^	I	T	T	T	T	Increased binding to mammal-like receptor (H5N1, H9N2)	[57,58]
	S159N	N	N	N	N	N	Increased binding to mammal-like receptor (H5N1)	[59]
	D225G	G	G	G	G	G	Increased transmission and replication in swine (H1N1 backbone with HA and NA of H9N2)	[60]
NA	A30T ^c^	I	I	T	T	T	Observed in airborne transmission in ferrets (H9N2)	[61]
	I117T	I	T	T	T	T	Reduced susceptibility to oseltamivir and zanamivir (H5N1)	[62]
NS1	P42S ^a^	S	S	S	S	S	Increased virulence and decreased antiviral response in mice (H5N1)	[63]
M1	N30D ^a^	D	D	D	D	D	Increased virulence in mice (H5N1)	[64]

* Not applicable (n.a.) due to insufficient sequence coverage. ^a^ Mutations/motifs are numbered according to alignments with A/Goose/Guangdong/1/1996 (H5N1). ^b^ Mutations/motifs are H3 numbered according to alignments with A/Aichi/2/1968 (H3N2). ^c^ Mutations/motifs are N2 numbered according to alignments with A/Aichi/2/1968 (H3N2).

**Table 4 animals-14-01018-t004:** Highest nucleotide sequence similarities to the AIVs obtained in the current study according to BLAST search tool on the GISAID Epi-Flu Database.

AIV Strain	Gene Segment	Highest Homology AIV Strain	Accession Number	% Homology
A/teal/Italy/1821-10_22VIR4622-1/2021 (H1N1)	PB2	A/*Anas_platyrhynchos*/Belgium/1098_2/2020(H1N1)	EPI1942951	99
	PB1	A/*Anas_platyrhynchos*/Belgium/2213_0006/2021(H11N6)	EPI2122874	99
	PA	A/*Anas_platyrhynchos*/Belgium/1098_2/2020(H1N1)	EPI1942953	98
	HA	A/*Anas_platyrhynchos*/Belgium/1098_2/2020(H1N1)	EPI1942954	99
	NP	A/*Anas platyrhynchos*/Belgium/11025_44/2017 (H11N1)	EPI1774315	99
	NA	A/chicken/Denmark/S02750-3/2020(H5N1)	EPI1694135	98
	M	A/duck/Moscow/5881/2021(H3N2)	EPI2175875	99
	NS	A/Environment/Jiangxi/12590/2019 (H10N3)	EPI1848446	99
A/mallard/Italy/22VIR4203-2/2021 (H5N3)	PB2	A/teal/Italy/21VIR8923-73/2021(H5N3)	EPI7987343	99
	PB1	A/duck/Italy/21VIR8022-1/2021(H5N3)	EPI1946718	99
	PA	A/duck/Italy/21VIR8022-1/2021(H5N3)	EPI1946719	99
	HA	A/duck/Italy/21VIR8022-1/2021(H5N3)	EPI1946720	99
	NP	A/duck/Italy/21VIR8022-1/2021(H5N3)	EPI1946721	99
	NA	A/duck/Italy/21VIR8022-1/2021(H5N3)	EPI1946726	99
	M	A/teal/Italy/21VIR8923-73/2021(H5N3)	EPI1947344	99
	NS	A/duck/Italy/21VIR8022-1/2021 (H5N3)	EPI1946723	99
A/teal/Italy/1856-7_22VIR4622-7/2021 (H9N2); A/teal/Italy/1828-6_22VIR4622-5/2021 (H9N2); A/teal/Italy/1821-14_22VIR4622-3/2021 (H9N2).	PB2	A/*Anas platyrhynchos*/Belgium/10402_H195386/2017 (H1N1)	EPI1775505	98
	PB1	A/duck/Italy/21VIR8024-4/2021(H5N3)	EPI1947293	99
	PA	A/shoveler/Novosibirsk region/3465k/2020 (H3N8)	EPI1849977	98
	HA	A/mallard/Poland/P096/2020(H9N2)	EPI2618292	98
	NP	A/Mallard/Netherlands/37/2015(H3N8)	EPI1530590	98
	NA	A/environment/England/030642/2020 (H5N2)	EPI2062134	98
	M	A/pintail/Egypt/MB-D-384C/2015(H3N6)	EPI1581277	98
	NS	A/mallard duck/Netherlands/41/2015 (H5N1)	EPI1306985	98

**Table 5 animals-14-01018-t005:** Amino acid substitutions of the HA and NA gene segments of AIVs isolated in the current study in comparison with the highest homologous strains obtained by BLAST search tool on GISAID Epi-Flu database.

Gene	AIV Strain		Subtype		Amino Acid Substitutions *	Source	
	This Study	Best Similar	This Study	Best Similar		This Study	Best Similar
HA	A/teal/Italy/1821-10_22VIR4622-1/2021	A/*Anas_platyrhynchos*/ Belgium/1098_2/2020	H1N1	H1N1	456 (W → R)	Eurasian teal, Bologna province, 15 October 2021	Mallard, Belgium, 12 September 2020
NA	A/teal/Italy/ 1821-10_22VIR4622-1/2021	A/chicken/Denmark/ S02750-3/2020	H1N1	H5N1 (LPAIV)	79 (I → V) 83 (T → A) 283 (M → I)	Eurasian teal, Bologna province, 15 October 2021	Chicken, Denmark, 28 January 2020
HA	A/mallard/Italy/ 22VIR4203-2/2021	A/duck/Italy/ 21VIR8022-1/2021	H5N3 (LPAIV)	H5N3 (LPAIV)	81 (K → N) 395 (I → V)	Mallard, Emilia-Romagna region, 13 October 2021	Duck, Italy, Lombardy region, 23 September 2021
NA	A/mallard/Italy/ 22VIR4203-2/2021	A/duck/Italy/ 21VIR8022-1/2021	H5N3 (LPAIV)	H5N3 (LPAIV)	169 (V → I)	Mallard, Emilia-Romagna region, 13 October 2021	Duck; Italy, Lombardy region; 23 September 2021
HA	A/teal/Italy/1856-7_22VIR4622-7/2021(H9N2) °	A/mallard/Poland/ P096/2020	H9N2	H9N2	245 (I → V) 419 (D → N)	Eurasian teal; Bologna province, October–November 2021	Duck; Poland; 2 of November 2020
NA	A/teal/Italy/1856-7_22VIR4622-7/2021(H9N2) °	A/environment/England/ 030642/2020	H9N2	H5N2 (LPAIV)	43 (S → N) 216 (G → V) 262 (V → I) 302 (I → V) 313 (D → G)	Eurasian teal, Bologna province, October–November 2021	Faeces, England, 31 of October 2020

° The HA and NA gene segments of A/teal/Italy/1828-6_22VIR4622-5/2021(H9N2) and A/teal/Italy/1821-14_22VIR4622-3/2021(H9N2) were identical. * H3 amino acid numbering; N2 amino acid numbering.

## Data Availability

The consensus sequences of the viruses analyzed in this study were submitted to the GISAID EpiFlu™ database under the accession numbers reported in Appendix A.

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
