# Peer review of "Genetic Diversity of Avian Influenza Viruses Detected in Waterbirds in Northeast Italy Using Two Different Sampling Strategies"

_animals, 2024, doi:10.3390/ani14071018_

Round 1

Reviewer 1 Report

Comments and Suggestions for Authors

I would suggest changing the title to using “two different sampling strategies”

52-please add “potentially” evolve into highly pathogenic

In the methodology, was the sampling of fecal droppings performed in regions exposed to sunlight or in the shade? Please mention in the methodology as this would affect the temperature of the droppings and UV light could potentially inactivate the viruses

What do the authors postulate for the low level of detection of influenza viruses in the

Charadriiformes order from both Cs and FD? As these are known to harbor low pathogenic avian influenza viruses naturally. Please expand on this in  the discussion

-How did the authors ensure that there was not cross contamination of feces in the environmental sampling techniques, for example leaching of feces from one species onto the feces of another bird

-Could the authors expand on the morphological characteristics of the feces types for instance in a supplementary figure or so

116-117 does other environmental factors such as humidity, water salinity and pH of the soil affect the survivability of the virus, please mention these factors in the discussion as well.

171-Please explain what is meant by mammal-like receptors, are you referring to the sialic acid repertoire

190—191 please replace wild birds with wild water birds

In the discussion, please clarify as the authors state that the shorebirds had the highest positivity rate, however in the results the ducks are stated to have the highest positivity rate

Comments on the Quality of English Language

Minor grammatical editing needed.

Reviewer 2 Report

Comments and Suggestions for Authors

During very critical period for AIV epidemiological monitoring, the authors did the sampling and tests with samples collected from wild birds in Northeast Italy from 2021 to 2022, and the authors showed that how viral population had been during that period at the very important wild bird habitats.

Some results need to be explained more and interpreted again;

1. Among 129 birds from which the CSs were collected , how many birds were hunted or found dead at what month? This information needs to be included in the article, and its epidemiological meaning in relation to viral detection rate needs to be described.

2. For further analysis, for example, the effect of the factors, male/female, adult/juvenile, dead/hunted on the virus detection and isolation rate may be analysed and added, hopefully.

3. It is believed that the authors subtyped the viruses with the genomes in the samples directly with rRT-PCR. It needs to be described whether the subtypes determined from sequencing on the isolates does match the ones from raw samples; there may be discrepancies sometimes. 

4. The authors said there were three inland wetlands for sampling, but in the manuscript there are only two inland wetlands designated. Actually, the inforamtion on how many samples collected and how many viruses are detected from each the different wetlands are absent.

5. The authors considered "the Campotto-Bassarone wetland" to be important area for breeding populations. One virus, H5N3, appears to be isolated this site located in Ferrara province on October or November, 2021 (Table S1). Any information on this wetland as a wintering site may need to be described.

6. I think the major limitation of this study is the absence of  oropharyngeal swab samples from the birds dead or hunted because in general the detection rate for HPAIV is higher in OP swabs than in CSs. For this, the authors need to explain.

Comments on the Quality of English Language

1. "found death wild birds" may need to be replaced by "found dead wild birds" 

2. "FDs samplings" need to be corrected. FDs is right?

3. graylag goose -> greylag goose
